# A genome-wide association study identifies 5 loci associated with frozen shoulder and implicates diabetes as a causal risk factor

Harry D. Green[1‡], Alistair Jones[2‡], Jonathan P. Evans[2], Andrew R. Wood[1], Robin N. Beaumont[1], Jessica Tyrrell[1], Timothy M. Frayling[1], Christopher Smith[2‡], Michael N. Weedon[2‡]*

1 Genetics of Complex Traits, University of Exeter Medical School, Exeter, United Kingdom, 2 Shoulder Unit, Princess Elizabeth Orthopaedic Centre, Royal Devon and Exeter Hospital, Exeter, United Kingdom

‡ HDG and AJ share equal first authorship on this work. CS and MNW share joint last authorship on this work.
* m.n.weedon@exeter.ac.uk

**Funding:** HG was funded by an "Expanding excellence in England" award from Research England. TMF has received funding from the Medical Research Council, MR/T002239/1 and the

## Abstract

Frozen shoulder is a painful condition that often requires surgery and affects up to 5% of individuals aged 40–60 years. Little is known about the causes of the condition, but diabetes is a strong risk factor. To begin to understand the biological mechanisms involved, we aimed to identify genetic variants associated with frozen shoulder and to use Mendelian randomization to test the causal role of diabetes. We performed a genome-wide association study (GWAS) of frozen shoulder in the UK Biobank using data from 10,104 cases identified from inpatient, surgical and primary care codes. We used data from FinnGen for replication and meta-analysis. We used one-sample and two-sample Mendelian randomization approaches to test for a causal association of diabetes with frozen shoulder. We identified five genome-wide significant loci. The most significant locus (lead SNP rs28971325; OR = 1.20, [95% CI: 1.16–1.24], p = 5x10$^{-29}$) contained *WNT7B*. This variant was also associated with Dupuytren's disease (OR = 2.31 [2.24, 2.39], p<1x10$^{-300}$) as were a further two of the frozen shoulder associated variants. The Mendelian randomization results provided evidence that type 1 diabetes is a causal risk factor for frozen shoulder (OR = 1.03 [1.02–1.05], p = 3x10$^{-6}$). There was no evidence that obesity was causally associated with frozen shoulder, suggesting that diabetes influences risk of the condition through glycemic rather than mechanical effects. We have identified genetic loci associated with frozen shoulder. There is a large overlap with Dupuytren's disease associated loci. Diabetes is a likely causal risk factor. Our results provide evidence of biological mechanisms involved in this common painful condition.

## Author summary

Frozen shoulder is a painful condition that often requires surgery and affects up to 5% of individuals aged 40–60 years. Little is known about the causes but it is known to be more common in people with diabetes. In a dataset of 500,000 people, we used a genome-wide

Innovative Medicines Initiative 2 Joint Undertaking under grant agreement No 875534. This Joint Undertaking support from the European Union's Horizon 2020 research and innovation programme and EFPIA and T1D Exchange, JDRF, and Obesity Action Coalition. The funders had no role in study design, data collection and analysis, decision to publish, or preparation of the manuscript.

**Competing interests:** The authors have declared that no competing interests exist.

association study to find genetic causes and a genetic technique called Mendelian randomisation to test if diabetes causes frozen shoulder. We found five new genetic variants that associate with frozen shoulder, and showed genetic overlap with Dupuytren's Disease, a similar condition that affects the fingers. We found an association with diabetes and obesity, but that the obesity association disappeared when we accounted for diabetes status, suggesting the condition is glycaemic rather than mechanical. Our Mendelian randomisation study showed evidence that type 1 diabetes has a causal effect on development of frozen shoulder, likely through a pathway involving long-term high blood glucose levels.

## Introduction

Frozen shoulder, also known as adhesive capsulitis, affects 2–5% of the population at some point in their lives [1,2]. It is characterised by initial shoulder pain followed by a gradual reduction in range of movement or "freezing". Pain subsides but joint stiffness can continue for years, causing significant disability [3]. The onset of frozen shoulder is usually between 40 and 60 years [4].

Little is known about the causes of frozen shoulder. Genome-wide association studies can provide new insights into underlying biological mechanism and possible drug targets. For example, a recent GWAS of osteoarthritis has been used to identify new therapeutic targets for the condition [5]. To date, there have been no published genome wide association studies (GWAS) of frozen shoulder.

Diabetes is the strongest known risk factor for frozen shoulder. Individuals with diabetes have a greatly increased lifetime risk, with a hazard ratio of 1.33 [6]. Whether diabetes causes frozen shoulder is unclear because the association may reflect residual confounding by other risk factors such as age, obesity [7] and Dupuytren's disease [8].

Mendelian randomization is a statistical method that can be used to infer causal relationships between an exposure and an outcome by using genetic variants associated with the exposure [9]. The exposure associated variants can be used as an unconfounded proxy for the exposure, as their inheritance is random at conception. This method is now extensively used to infer causal associations and proof of principle examples include evidence that increased BMI causes diabetes [10] and increased LDL cholesterol causes coronary artery disease [11].

Using the UK Biobank, we performed a genome-wide association study (GWAS) to identify genetic variants associated with frozen shoulder. We used publicly available summary statistics from 176,899 individuals from the FinnGen study for replication [12]. We then used Mendelian randomization to test for causal associations of diabetes and obesity with frozen shoulder.

## Methods

### Ethics statement

Ethics approval for the UK Biobank study was obtained from the North West Centre for Research Ethics Committee (11/NW/0382)[13]. Written informed consent was obtained from all participants.

### Frozen shoulder cases in the UK Biobank

The primary GWAS was performed on cases of frozen shoulder identified from the ICD-10 code M750, OPCS4 code W871 and read codes (N210, XE1FL and XE1Hm) from the primary

care data in UK Biobank. Controls were defined as individuals without a record of one of these codes.

## Other phenotypes

ICD-10 codes were used for related disorders (Dupuytren's disease (M720), rotator cuff (M751), and calcific tendinitis of shoulder (M753)). Further, OPCS in Supplementary Table 1 of [14] (all sections except anatomical codes) and primary care read codes 7H320, 7H340, N236., Xa8RB, XE07l, XE1Fj, XM0ug, and XM1HO were used for Dupuytren's disease. Type 1 diabetes was classified favouring specificity over sensitivity, defined by diagnosis aged $\leq$ 20, on insulin within 1 year of diagnosis and at time of recruitment, not using oral antihyperglycemic agents, and not self-reporting type 2 diabetes, as described in [15]. Type 2 diabetes was defined by participants answering yes to the question 'Has a doctor ever told you that you have diabetes?' excluding those who reported using insulin within one year of diagnosis, were diagnosed under the age of 35, or were diagnosed within the past year [16]. We used UK Biobank variable 21001 for BMI measurement. HbA1c in mmol/mol is defined using variable n_30750_0_0 in UK Biobank. We defined diabetic retinopathy using self-report code 1276, diabetic neuropathy / ulcers using 1468 and diabetic nephropathy using 1607. Diabetes duration was defined by the difference between age at baseline and data-field 2976 (age diabetes diagnosed).

## Genome-wide association study

We performed two case-control genome-wide association studies using BOLT-LMM [17], which applies a linear mixed model to test for an association between each SNP and the outcome trait, including population structure as a part of the model. Age, sex, study centre and genotyping chip were also included as covariates. We performed our primary GWAS using ICD10, OPCS and primary care codes, and a sensitivity analysis including only cases identified in ICD10 and OPCS. Briefly, principal component analysis was performed using individuals from the 1000 Genomes Project prior to projection of UK Biobank individuals into the principal component space. K-means clustering was subsequently applied to classify individuals as European, with centres initiated to the mean principal component values of each 1000 Genomes sub-population. The first 4 principal components were used in this analysis. We used LocusZoom to plot the resulting significant locus [18] and GTEx V8 to explore the possibility of functional variants [19]. P-values $< 5\mathrm{x}10^{-8}$ were considered significant for the GWAS. Betas and standard errors from BOLT-LMM were converted to log-odds ratios using $\log(OR) = \beta/(\mu(1-\mu))$, where $\mu$ is the case-control ratio, as per the BOLT-LMM user manual [17]. Standard errors were also divided by $\mu(1-\mu)$.

## Replication and meta-analysis

We used summary statistics from freeze 4 of the FinnGen publicly available resource [12] to replicate the genome-wide significant findings in the discovery GWAS. Cases of frozen shoulder were defined in FinnGen using the ICD-10 code M750. METAL was used to perform a genome-wide meta-analysis of UKBB and FinnGen results, based on log-OR and standard errors [20]. We performed two meta-analyses using UK Biobank and FinnGen, the first using all cases from UK Biobank and the second using ICD-10 and OPCS codes.

## Observational analyses

Observational analyses were performed on a subset of 379,708 unrelated individuals, 2,132 cases and 377,576 controls. We used a KING Kinship[21] to exclude those third-degree

relatives or closer. An optimal list of unrelated individuals was generated by preferentially removing individuals with the maximum number of relatives to allow maximum numbers of individuals to be included.

We tested for associations with demographic and clinical features using logistic regression models, firstly using a univariable logistic regression model on 7 variables of interest (sex, age, TDI, type 1 diabetes, type 2 diabetes, BMI and WHR) and secondly using all variables in a multivariable logistic regression model to identify independent associations. We analysed Type 1 and Type 2 diabetes separately. $p < 0.007$ (0.05/7) was considered significant. We performed a separate analysis specifically of diabetes-related variables (type, duration, retinopathy, neuropathy and nephropathy) for which $p < 0.01$ (0.05/5) was considered significant.

## Mendelian randomization

We applied two different methods. The one sample Mendelian randomization (MR) results were performed in two stages: first, the association between each exposure and a genetic risk score was used to derive a genetically predicted exposure value, and second, these predicted values were used in a logistic regression model of frozen shoulder, adjusting for ancestral principal components, assessment centre, genotyping platform, age and sex.

Second, we used two-sample methods, which involve regressing the effect sizes of variant–outcome associations against the effect sizes of the variant–risk factor associations for a set of SNPs associated with the exposure. We performed inverse variance weighted (IVW) instrumental variable analysis and two further methods that are more robust to potential violations of the standard MR assumptions (MR-Egger [22] and weighted-median MR [23]). We used effect sizes and 30 SNPs for type 1 diabetes reported in [24] using the Wellcome Trust Case Control Consortium, and a type 2 diabetes genetic risk score of 88 variants identified in [25,26]. The T1D-GRS uses 2 SNPs, rs2187668 and rs7454108, to tag and categorize the high risk DR3/DR4 haplotypes [24], this requires access to individual level data which we did not have for the FinnGen study. We therefore performed two analyses: one where we used the 2 SNPs to impute DR3/DR4 status and another where we used the SNPs as standard risk SNPs. We performed the above two-sample Mendelian randomisation in UK Biobank cases identified by ICD10 and OPCS codes, all UK Biobank cases and the FinnGen cohort. We also performed Mendelian-Randomisation using log-odds ratios and standard errors from the two genome-wide meta-analyses.

P-values $< 0.01$ were considered significant for Mendelian randomisation results. Mendelian randomisation results using a diabetes status as an exposure should be interpreted in terms of liability, which may cause frozen shoulder via changes in diabetes status and / or by independent pathways such as glycaemia[27].

## Results

### Diabetes is associated with frozen shoulder in the UK Biobank

To identify associated risk factors for frozen shoulder we tested several measures of diabetes and its related traits to assess their independent contributions. There were 2,540 cases of frozen shoulder in the UK Biobank based on ICD-10 and OPCS codes, and 10,104 including GP records. A Venn diagram showing overlap can be found in **S1 Fig**. Demographic and clinical associations with frozen shoulder defined by ICD10 and OPCS can be found in **Table 1**. Those with frozen shoulder were more likely female, had lower Townsend deprivation index, more likely obese (higher BMI and waist-hip-ratio) and more likely to have either type 1 or type 2 diabetes, with type 1 giving the strongest association. In a multivariable logistic regression model, only sex, types 1 and type 2 diabetes showed association.

**Table 1. Demographic and clinical associations with frozen shoulder in the UK Biobank.** The OR and p value columns were calculated using univariable logistic regression. The adjusted OR and adjusted p value columns were calculated using a multivariable logistic regression model. Adjusted results for diabetes types were calculated from a model excluding the other type. These analyses used only unrelated individuals in the UK Biobank.

| Variable | Cases (2,132) | Controls (377,576) | OR [95% CI] | p value | Adjusted OR [95% CI] | Adjusted p value |
|---|---|---|---|---|---|---|
| Male sex | 174,125 (46%) | 820 (38%) | 0.73 [0.67–0.80] | $2\times10^{-12}$ | 0.64 [0.58–0.71] | $1\times10^{-18}$ |
| Age [SD] | 57.2 [8.0] | 57.2 [7.4] | 1.00 [0.99–1.01] | 0.86 | 1 [1.00–1.00] | 0.59 |
| Townsend deprivation index [SD] | -1.26 [3.1] | -1.48 [3.0] | 1.02 [1.01–1.04] | $7\times10^{-4}$ | 1.01 [0.99–1.03] | 0.10 |
| Type 1 diabetes | 23 (1.2%) | 356 (0.09%) | 12.23 [8.00–18.69] | $9\times10^{-31}$ | 13.09 [8.55–20.03] | $2\times10^{-32}$ |
| Type 2 diabetes | 179 (9.0%) | 11,887 (3.2%) | 2.99 [2.56–3.49] | $1\times10^{-43}$ | 3.00 [2.54–3.54] | $6\times10^{-39}$ |
| BMI [SD] | 28.0 [4.9] | 27.4 [4.8] | 1.12 [1.07–1.18] | $1\times10^{-10}$ | 1.03 [0.97–1.09] | 0.12 |
| WHR [SD] | 0.873 [0.09] | 0.871 [0.09] | 1.16 [1.11–1.21] | $1\times10^{-17}$ | 1.03 [0.97–1.09] | 0.09 |

Diabetes duration was nominally associated with frozen shoulder adjusted for diabetes type (OR per additional year with diabetes 1.02 [1.00–1.04], p = 0.02). HbA1c associated with frozen shoulder independently of type or duration of diabetes (OR = 1.09 [95% CI: 1.02–1.17], p = 0.009). Diabetic retinopathy (OR = 1.99 [95% CI, 1.19–3.33], p = 0.009) associated with frozen shoulder independently of type or duration of diabetes. This suggests that individuals with longer duration and less well controlled diabetes have higher risk of frozen shoulder.

## Three loci are associated with frozen shoulder from discovery GWAS

**Fig 1** presents the results of the frozen shoulder genome-wide association study using all frozen shoulder cases. We identify a genome-wide significant peak on chromosome 22. The A allele at the lead SNP, rs28971325, has a frequency of 26.4% in cases and 23.2% in controls (OR = 1.20, [95% CI: 1.16–1.24], p = $5\times10^{-29}$). We also observe two additional genome-wide

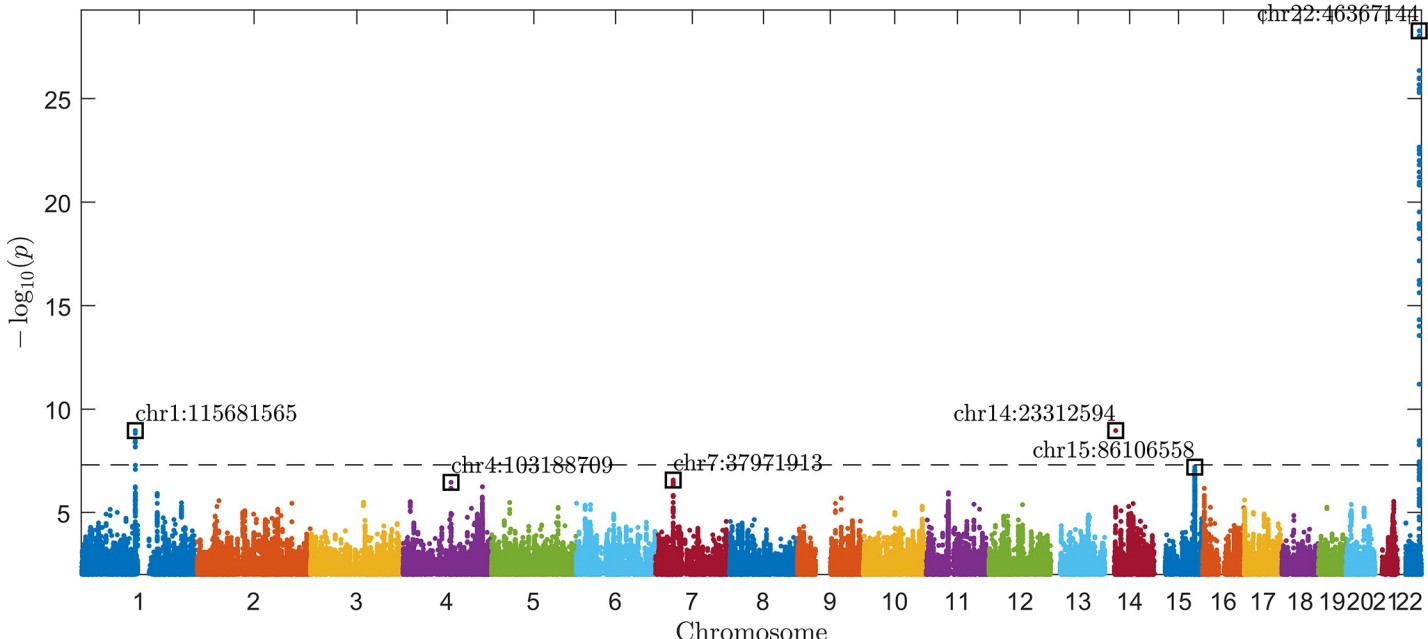

**Fig 1. Manhattan plot of discovery GWAS for frozen shoulder in UK Biobank.** The plot shows –log10(p) values for each single nucleotide polymorphism [SNP] in the HRC Imputation Panel and their association with frozen shoulder defined by ICD10, OPCS codes and GP records with p < 0.01, computed using BOLT-LMM and plotted using an in-house MATLAB script which we have made publicly available on the MATLAB File Exchange [31]. The horizontal dashed line is the genome-wide significance threshold at p = $5\times10^{-8}$. Positions are based on the hg19 reference human genome.

significant signals: rs5777216 on chromosome 1 (OR = 0.92 [95%CI 0.89–0.94], p = 1x10$^{-9}$) and rs1042704 on chromosome 14 (OR = 1.11 [95%CI 1.07–1.15], p = 1x10$^{-9}$).

When including only ICD10 and OPCS codes in the analysis, rs28971325 associated with OR = 1.32 (95%CI 1.22–1.41), p = 7x10$^{-17}$. rs5777216 (OR = 0.91 [95%CI 0.86–0.96], p = 1x10$^{-3}$) and rs1042704 (OR = 1.12 [95%CI 1.05–1.20], p = 8x10$^{-4}$) were not genome-wide significant. A Manhattan Plot showing these results can be found in S2 Fig.

## Replication and meta-analyses of genome-wide association studies

We sought replication of the association signals in the FinnGen study. FinnGen have recently released GWAS summary statistics for 1801 diseases and traits, including frozen shoulder [12]. The lead SNP from the UK Biobank, rs28971325, was associated with frozen shoulder in Finn-Gen with a similar effect size to the UK Biobank results using ICD10 and OPCS codes (OR = 1.30 [95%CI 1.20–1.41], p = 6x10$^{-10}$). The frequency of the A allele was 20.8% in cases compared to 17.6% in controls. rs1042704 was nominally associated with frozen shoulder in the FinnGen cohort (OR = 1.15 [95%CI: 1.07–1.24] p = 2x10$^{-4}$), while rs5777216 (OR = 0.94 [95%CI: 0.89–1.00] p = 6x10$^{-2}$) was not.

We note five genome-wide significant loci when performing a meta-analysis using all UK Biobank cases and FinnGen, including the three previously associated loci. Odds ratios for these SNPs were similar across all three definitions, except for rs28971325, which was weaker in UKBB's GP records. The summary statistics can be found in Table 2. Manhattan Plots of the meta-analyses can be found in S3 and S4 Figs.

## WNT7B, MMP14 and SFRP4 are potential causal genes

S5–S9 Figs provide LocusZoom plots [18] for the 5 associated loci. For the strongest associated locus, rs28971325 (S5 Fig), there were no coding variants with an r$^2$ > 0.8. There are no gene expression associations in GTEX for rs28971325. However, a recent study found *WNT7B* was the second most differentially expressed transcript genome-wide with a log fold change of 7 (*p* = 1x10$^{-16}$) in anterior capsule tissue from 22 patients undergoing arthroscopic capsulotomy surgery for frozen shoulder compared to 26 controls [28]. The lead SNP at the next strongest associated locus, rs1042704, is a missense variant in the *MMP14* gene (S6 Fig). This variant has previously been associated with Dupuytren's disease and has been shown to have a 99% posterior probability of being the causal variant [29]. The chromosome 7 locus is also a known Dupuytren's locus, and previous work has suggested *SFRP4* is the causal gene [29].

## Frozen shoulder associated SNPs have been associated with Dupuytren's contracture and bone mineral density

A previous GWAS study of Dupuytren's disease found an association in the same genomic region of *WNT7B* [29]. The lead SNP for Dupuytren's disease, rs7291412, was not associated

**Table 2. Association of lead SNPs in the full meta analysis in the two input GWAS results (ICD+OPCS+GP in UKBB and FinnGen) and the UKBB result excluding GP records.** The effect allele frequency is from the UK Biobank cohort.

| SNP | Effect Allele (Freq) | Meta Analysis | ICD+OPCS | ICD+OPCS+GP | FinnGen |
|---|---|---|---|---|---|
| 22:46367144 rs28971325 | A (0.233) | 1.21 (1.25–1.18) 8E-37 | 1.32 (1.24–1.41) p = 7E-17 | 1.20 (1.16–1.24) p = 5E-29 | 1.30 (1.20–1.41) p = 6E-10 |
| 14:23312594 rs1042704 | A (0.215) | 1.12 (1.15–1.08) 1E-12 | 1.12 (1.05–1.20) p = 8E-04 | 1.11 (1.07–1.15) p = 1E-09 | 1.15 (1.07–1.24) p = 2E-04 |
| 1:115681565 rs5777216 | GT (0.403) | 1.09 (1.11–1.06) 3E-10 | 0.91 (0.86–0.96) p = 1E-03 | 0.92 (0.89–0.94) p = 1E-09 | 0.94 (0.89–1.00) p = 6E-02 |
| 15:86125607 rs17570529 | T (0.266) | 0.92 (0.94–0.89) 3E-09 | 1.10 (1.03–1.17) p = 4E-03 | 1.09 (1.05–1.12) p = 1E-07 | 1.10 (1.03–1.18) p = 6E-03 |
| 7:37981961 rs2472660 | C (0.355) | 0.93 (0.95–0.90) 2E-08 | 1.06 (1.00–1.12) p = 4E-02 | 1.07 (1.04–1.10) p = 6E-06 | 1.13 (1.06–1.20) p = 3E-04 |

**Table 3. Association of frozen shoulder genetic loci with Dupuytren's Disease.**

| CHR | SNP | BP | OR | 95%CI | P |
|-----|-----|-----|-----|-------|---|
| 22 | rs28971325 | 46367144 | 2.31 | 2.24–2.39 | $<1\times10^{-300}$ |
| 14 | rs1042704 | 23312594 | 1.18 | 1.14–1.22 | $3\times10^{-21}$ |
| 1 | rs5777216 | 115681565 | 1.02 | 0.99–1.05 | 0.21 |
| 15 | rs17570529 | 86125607 | 1.08 | 1.04–1.11 | $1\times10^{-5}$ |
| 7 | rs2472660 | 37981961 | 1.32 | 1.28–1.36 | $7\times10^{-73}$ |

with frozen shoulder in UK Biobank (OR = 0.98 [95%CI: 0.95–1.01], p = 0.17) or FinnGen (OR = 1.00 [95%: 0.94, 1.06], p = 0.92). Of the five genome-wide significant loci in the meta-analysis, rs28971325, rs1042704 and rs2472660 associated with Dupuytren's disease at the genome-wide significant level, rs17570529 was nominally significant (p = $1\times10^{-5}$), and rs2472660 did not associate (p = 0.21) (Table 3). None of these five loci associated with either rotator cuff or calcific tendinitis of shoulder at p<0.01.

## Dupuytren's disease and other fibroblastic diseases

There was limited overlap between Frozen Shoulder and Dupuytren's disease observationally: 61 (2.4%) of frozen shoulder cases in the ICD10+OPCS definition and 198 (2.0%) including GP record cases had an ICD10 code for Dupuytren's disease. However, many of the previously reported Dupuytren's disease SNPs were associated with frozen shoulder with a correlation in effect sizes ($R^2$ = 0.72, p = $1\times10^{-7}$, S10 Fig). The association of the 5 SNPs with frozen shoulder did not change when we excluded individuals with Dupuytren's disease from the cases.

## Mendelian randomization provides evidence that diabetes is causal to frozen shoulder

We used Mendelian randomization to explore whether the associations with Type 1 and Type 2 diabetes reported in Table 1 were causal. Using 1-sample Mendelian randomization methods, genetic data provided evidence that type 1 diabetes causes frozen shoulder: OR 1.05 [95% CI: 1.02–1.09], p = 0.002 using ICD-10 and OPCS codes and OR 1.04 [95% CI: 1.02–1.06], p = $2\times10^{-6}$ including primary care records. Evidence of a causal role of type 2 diabetes was weaker: OR 1.10 [95% CI: 0.99–1.22], p = 0.07 using ICD-10 codes and OR 1.07 [95% CI: 1.02–1.13], p = 0.006 including cases identified by UK Biobank primary care records. These results were consistent when we performed a sensitivity analysis adjusting both stages of the regression for significant variables in Table 1 (BMI, WHR and TDI) to correct for potential pleiotropy (S1 Table).

The association with type 1 diabetes was replicated using more robust, two-sample MR methods IVW (p = $3\times10^{-6}$) and MR-Egger (p = $2\times10^{-6}$). There was limited evidence from the two-sample methods that type 2 diabetes causes frozen shoulder (IVW: p = 0.06, MR-Egger: p = 0.96). The supplementary tables contain the full results from our Mendelian randomization analyses, including T1D-GRS results that exclude HLA variants which have greater potential for pleiotropy, for which the effect sizes were consistent (S1 Table shows the two stage regression, S2 Table shows the IVW results and heterogeneity p values, and S3 Table shows the MR-Egger results). Figs 2 and 3 show plots showing the lines of best fit for MR-Egger, IVW, Median IV and Penalised Median IV methods for type 1 and type 2 diabetes defined by ICD-10 codes, demonstrating in Fig 2: for type 1 diabetes SNPs, a strong correlation between the betas for type 1 diabetes and frozen shoulder and in Fig 3: no association for type 2 diabetes SNPs.

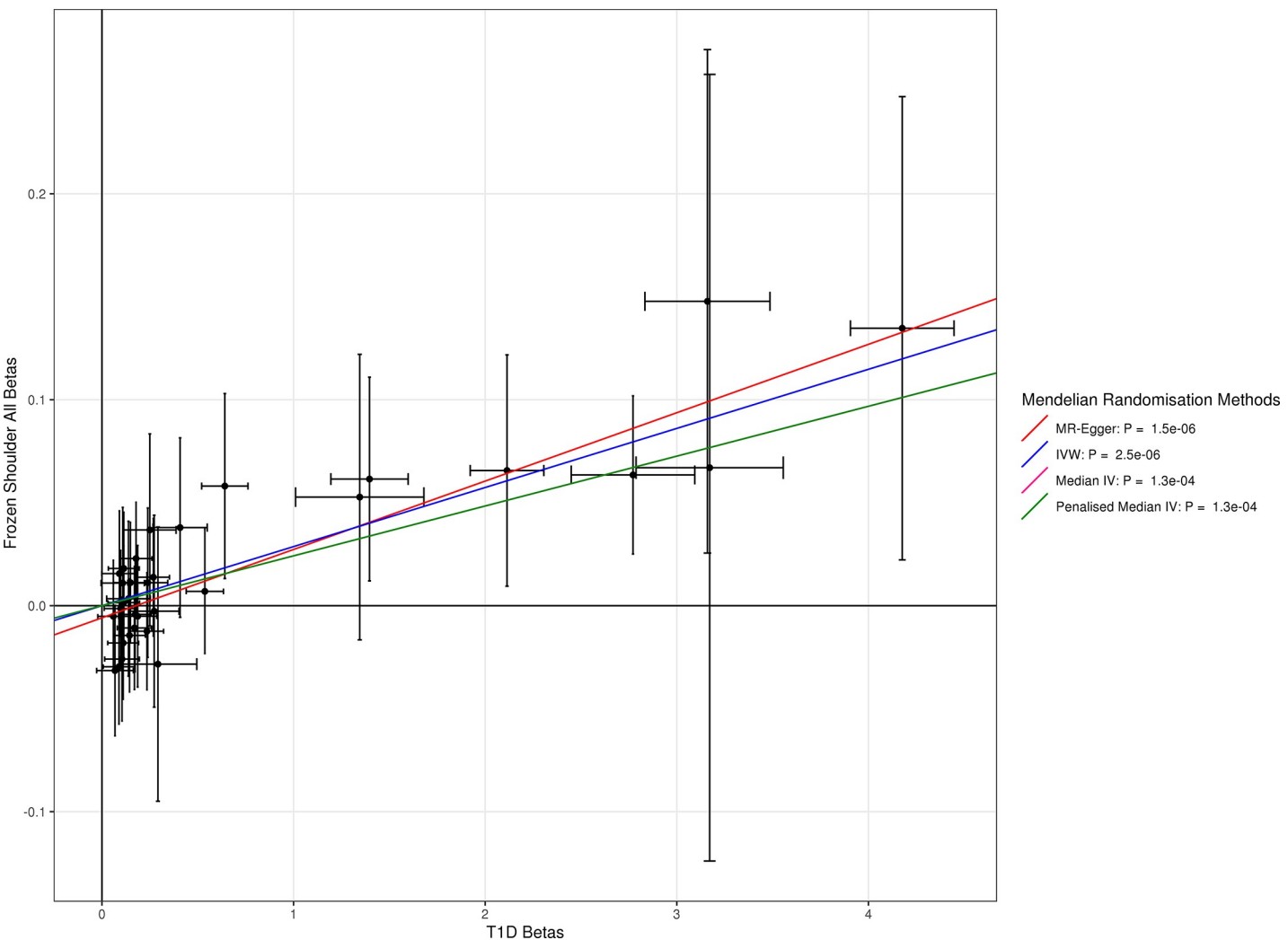

**Fig 2. Two Sample Mendelian Randomization Results, showing for each SNP in the Type 1 Diabetes GRS in [20], the log-odds ratio for type 1 diabetes on the x axis and the log-odds ratio for frozen shoulder (defined by ICD-10, OPCS and GP records) on the y axis.**

## Discussion

We have identified robust associations of common genetic variants with frozen shoulder. Frozen shoulder is a condition which affects up to 5% of the population around between the ages of 40 to 60, but the causes of the disease, and particularly the transient nature of the condition are unknown. The genome-wide association study and Mendelian randomization analyses we report here provide new insights into the underlying causes of the condition.

*WNT7B* is a candidate causal gene at the most strongly associated locus. As with most GWAS studies, further work is needed to identify the causal variant at the locus. GTEX analyses did not identify any strong eQTL's with the lead SNP at the associated locus. However, a recent study performed RNA-seq on anterior capsule tissue from 22 patients undergoing arthroscopic capsulotomy surgery for adhesive capsulitis and compared to 26 undergoing arthroscopic stabilization surgery for a different condition [28]. *WNT7B* was the second most differentially expressed transcript genome-wide with a log fold change of 7 ($p = 1 \times 10^{-16}$), although it was noted that the expression levels of *WNT7B* was relatively low. The WNT

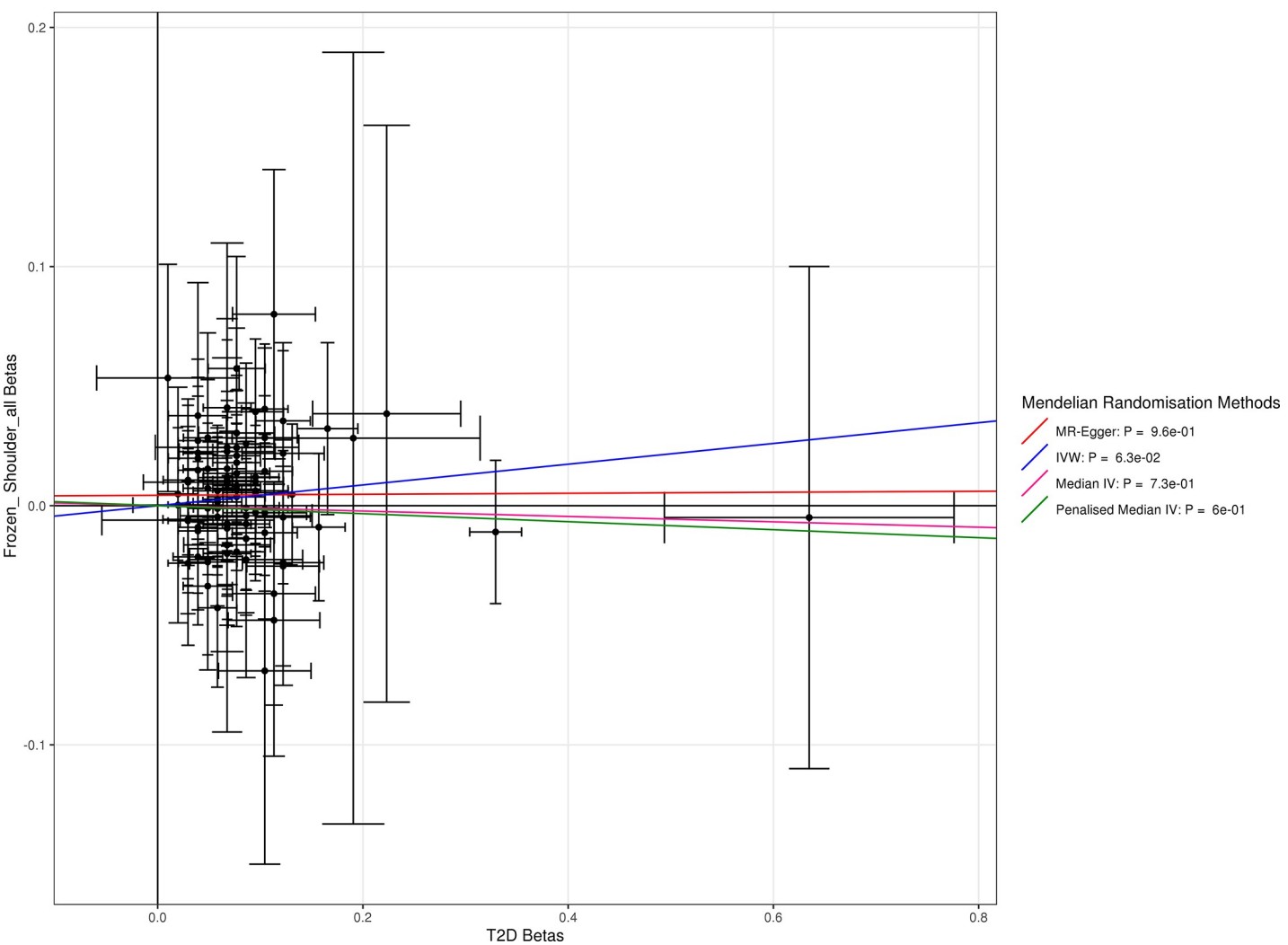

**Fig 3. Two Sample Mendelian Randomization Results, showing for each SNP in the Type 1 Diabetes GRS in [20], the log-odds ratio for type 2 diabetes on the x axis and the log-odds ratio for frozen shoulder (defined by ICD-10, OPCS and GP records) on the y axis.**

signalling pathway has been highlighted by previous GWAS of related fibroblastic diseases [30]. Other potential causal genes at associated loci included *MMP14*, a missense variant previously finemapped for Dupuytren's disease, and *SFRP4*.

Three of the identified loci for frozen shoulder also associate with Dupuytren's disease. Dupuytren's disease is a common condition which is characterised by a hand deformity where there is contracture of connective tissue within the palm and digits preventing full finger extension. A previous GWAS of Dupuytren's identified 26 associated loci with the condition [29]. One of the more strongly associated loci also contained the *WNT7B* locus. However, the lead SNP at that locus has not been associated with frozen shoulder and does not associate with frozen shoulder in the UK Biobank or FinnGen. The odds ratio for the lead SNP in our study has a significantly stronger odds ratio for Dupuytren's than the strongest association signal from the Dupuytren's GWAS. The explanation may be that older versions of SNP chips or imputation panels were used in the previous study which did not capture the lead SNP from our current study of frozen shoulder. Overall there is a strong overlap in signals between Dupuytren's disease and frozen shoulder. Some overlap might be expected, as both conditions

are a result of contracture of connective tissue planes. WNT is a known regulator of planar cell polarity and causes tissue planes to contract via the non-canonical pathway. Dupuytren's disease differs from frozen Shoulder in that frozen shoulder is most commonly a transient condition.

Diabetes is known to be an observational risk factor for frozen shoulder, but the causal nature of the association was unclear. Our Mendelian randomization analyses in UK Biobank provide evidence that Type 1 diabetes is causal for the condition. The weaker association with Type 2 diabetes is likely due to differences in duration of diabetes or earlier diabetes onset, as Type 1 diabetes is generally diagnosed earlier (half before the age of 30 years) whereas the diagnosis of Type 2 diabetes generally occurs later in life (after 50 years). It is likely all individuals with diabetes would have an increased risk of frozen shoulder, with those with longer duration (like those individuals with Type 1 diabetes) and worse glycaemic control at increased risk.

There are limitations to our study. The UK Biobank ICD-10 data is reliant on patients having a coded diagnosis at hospital, which is reliant on accuracy of hospital coding, and also reliant on the condition being severe enough for the patient to go to hospital. This may result in less serious cases being classified as controls. As such, they may be overly specific, while the primary care records are potentially overly sensitive and include cases that are not true frozen shoulder cases because it can be difficult to accurately diagnose in primary care. Consistent with this the effect size for frozen shoulder was smaller when cases were based on presence in primary care records rather than inpatient ICD-10 codes for the most associated loci. This study was performed using only white Europeans, and further study is needed to determine if results replicate for other ethnic groups. The UK Biobank only includes patients between the ages of 40 and 69 at recruitment, although this covers the most common incidence age range for frozen shoulder.

We have identified genetic loci associated with frozen shoulder. There is a large overlap with Dupuytren's disease genetic loci. Diabetes is a likely causal risk factor. Our results provide evidence of biological mechanisms involved in this common painful condition.

## Supporting information

**S1 Table. One sample Mendelian randomisation results.** A table showing results from a two-stage regression where the GRS for the exposure was regressed against the exposure, and the genetically predicted exposure was used to predict frozen shoulder. Adjusted refers to a sensitivity analysis in which both stages of the regression were adjusted for the significant variables in Table 1.
(DOCX)

**S2 Table. IVW Mendelian randomisation results.** A table showing results the IVW analysis. Meta Analysis 1 refers to using the betas and standard errors from the meta-analysis GWAS with FinnGen using ICD10 + OPCS from UKBB. Meta Analysis 2 refers to the same using ICD10 + OPCS + GP records from UKBB. The P het column contains heterogeneity.
(DOCX)

**S3 Table. MR-Egger results.** A table showing results the MR-Egger results. Meta Analysis 1 refers to using the betas and standard errors from the meta-analysis GWAS with FinnGen using ICD10 + OPCS from UKBB. Meta Analysis 2 refers to the same using ICD10 + OPCS + GP records from UKBB.
(DOCX)

**S1 Fig. Venn diagram of UK Biobank cases.** This Venn diagram shows overlap between the different case definitions of frozen shoulder (ICD10, OPCS, and GP record codes) in the UK

Biobank.
(TIF)

**S2 Fig. Manhattan plot of sensitivity analysis GWAS for frozen shoulder in UK Biobank.** The plot shows–log10(p) values for the association of each single nucleotide polymorphism [SNP] in the HRC Imputation Panel and their association with UK Biobank frozen shoulder cases defined by ICD10 and OPCS codes. The horizontal dashed line is the genome-wide significance threshold at p = 5×10–8.
(TIF)

**S3 Fig. Manhattan plot of primary meta-analysis.** The plot shows–log10(p) values for the association of each single nucleotide polymorphism [SNP] in the HRC Imputation Panel and their association in the meta-analysis using UK Biobank ICD10 + OPCS and FinnGen. The horizontal dashed line is the genome-wide significance threshold at p = 5×10–8.
(TIF)

**S4 Fig. Manhattan plot of secondary meta-analysis.** The plot shows–log10(p) values for the association of each single nucleotide polymorphism [SNP] in the HRC Imputation Panel and their association in the meta-analysis using UK Biobank ICD10 + OPCS + GP Records and FinnGen. The horizontal dashed line is the genome-wide significance threshold at p = 5×10–8.
(TIF)

**S5 Fig. LocusZoom plot of rs28971325.**
(TIF)

**S6 Fig. LocusZoom plot of rs1042704.**
(TIF)

**S7 Fig. LocusZoom plot of rs5777216.**
(TIF)

**S8 Fig. LocusZoom plot of rs17570529.**
(TIF)

**S9 Fig. LocusZoom plot of rs2472660.**
(TIF)

**S10 Fig. Scatterplot of Frozen Shoulder betas against Dupuytren betas for Dupuytrens SNPs.** Betas and 95% CIs for association between Frozen Shoulder and between Dupuytren's Disease for all Dupuytren's SNPs in Ng. et. al.
(TIF)

## Acknowledgments

This research has been conducted using the UK Biobank Resource (Application number: 9072). We want to acknowledge the participants and investigators of the FinnGen study. The authors acknowledge the use of the University of Exeter High-Performance Computing facility in carrying out this work.

## Author Contributions

**Conceptualization:** Alistair Jones, Jonathan P. Evans, Christopher Smith, Michael N. Weedon.

**Formal analysis:** Harry D. Green, Michael N. Weedon.

**Funding acquisition:** Timothy M. Frayling, Michael N. Weedon.

**Investigation:** Harry D. Green, Michael N. Weedon.

**Methodology:** Harry D. Green, Andrew R. Wood, Robin N. Beaumont, Jessica Tyrrell, Timothy M. Frayling, Michael N. Weedon.

**Software:** Harry D. Green, Andrew R. Wood, Robin N. Beaumont, Jessica Tyrrell, Michael N. Weedon.

**Supervision:** Timothy M. Frayling, Michael N. Weedon.

**Validation:** Harry D. Green.

**Visualization:** Harry D. Green.

**Writing – original draft:** Harry D. Green, Alistair Jones, Jonathan P. Evans, Andrew R. Wood, Robin N. Beaumont, Jessica Tyrrell, Timothy M. Frayling, Christopher Smith, Michael N. Weedon.

**Writing – review & editing:** Harry D. Green, Alistair Jones, Jonathan P. Evans, Andrew R. Wood, Robin N. Beaumont, Jessica Tyrrell, Timothy M. Frayling, Christopher Smith, Michael N. Weedon.

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
