## [Decision Letter · Decision Letter 0]

25 Jan 2021

Dear Dr Weedon,

Thank you very much for submitting your Research Article entitled 'A genome-wide association study of frozen shoulder identifies a common variant of WNT7B and diabetes as causal risk factors' to PLOS Genetics.

The manuscript was fully evaluated at the editorial level and by two peer reviewers. As you will see, both reviewers are generally positive but identify several major concerns that would be necessary to address for the manuscript to move forward. The comments of reviewer #2 should be straightforward to address, but note that the comments form reviewer #1 will require substantial effort. If the reviewers' comments can be addressed satisfactorily, we would be interested to evaluate a revised manuscript. We cannot, of course, promise publication at that time.

If you decide to revise the manuscript for further consideration at PLOS Genetics, please aim to resubmit within the next 60 days, unless it will take extra time to address the concerns of the reviewers, in which case we would appreciate an expected resubmission date by email to plosgenetics@plos.org.

[LINK]

We are sorry that we cannot be more positive about your manuscript at this stage. Please do not hesitate to contact us if you have any concerns or questions.

Yours sincerely,

Gregory S. Barsh

Editor-in-Chief

PLOS Genetics

Gregory Copenhaver

Editor-in-Chief

PLOS Genetics

Reviewer's Responses to Questions

**Comments to the Authors:**

Reviewer #1: PLOS Genetics peer review report

A genome-wide association study of frozen shoulder identifies a common variant of

WNT7B and diabetes as causal risk factors

Green et al.

This article describes a GWAS of frozen shoulder in UK Biobank with replication in FinnGen. The authors discovered 1 locus, and used MR to show that type 1 DM is a causal risk factor for frozen shoulder.

I enjoyed reading the paper. The work is novel, and will be of interest to the scientific community. The paper overall is well-written. I have suggested major changes that must be performed before acceptance, and would favour the authors being given a chance to perform these analyses and address my comments in a re-submission.

Major points

1. The authors have used only diagnostic codes to define disease cases for both FS and DD. This seriously underestimates the number of cases, as many are only identified by surgical (OPCS) codes. For example, for FS, a quick browse of the UK biobank data showcase reveals the following numbers of cases:

Code Cases

ICD10 – M750 2314

OPCS – W781 922

This misclassification may have led to a critical loss of power to detect association (I note several sub-threshold loci in figure 1). This is a missed opportunity, and should be rectified before publication.

Surgical codes for DD can be found in supplementary data file of https://doi.org/10.1038/s41598-020-73595-y

2. In a similar vein, GP data has been recently released, and the authors identify 7,913 cases of FS in the GP data. These cases are also misclassified in the GWAS analysis, and should be included as cases (note only 323 overlapped).

The authors looked only at the associated variant in this data I think, though it is not clear as there is no methodology reported for this part of the genetic analysis. The authors should combine the cases from both hospital records above and GP records into a proper case-control cohort to avoid misclassification and maximise power. They should then do a single proper GWAS. This again is a major missed opportunity to more accurately define the genetic architecture of FS, and should be performed before publication.

3. The reporting of the analysis in FinnGen is inadequate. Did the authors just test the single associated variant? What about suggestive variants that were below the genomewide significance threshold? Did they do a GWAS and then meta-analysis with the summary statistics? They should perform a full GWAS (or use the pre-prepared GWAS data) and meta-analysis as it is possible that this will reveal further associated loci, and further delineate the genetic architecture of FS. Again, a missed opportunity that should be rectified before publication.

4. Supplementary tables 2 and 3: I am unsure as to the precise definition of cases and controls here, and what the methodology for “meta-analysis” was (this is not described in the methods, and definitely needs to be), but I have a serious concern about performing meta-analysis when cases and (especially) controls are likely to be the same in two analyses. Specifically here in the “UK Biobank ICD” and “UK Biobank GP” cohorts I think it is likely that there is very large sample overlap. How is this dealt with in such a “meta-analysis”.

Minor points

1. Introduction Page 3 “Diabetes is the strongest known risk factor for Diabetes” – I think you mean for FS.

2. Remove “a deeply phenotyped…..from the UK” – PLOS genetics readers know the UK Biobank.

3. Results: “…had a lower Townsend Deprivation Index…” Not really. This is very marginally significant (if you were to Bonferroni correct your seven univariable tests), and is not significant in the multivariable model. Please remove.

4. “Diabetic eye disease….” – how many variables related to diabetes did you test? Did you do a Bonferroni correction? This is very marginally “significant” (p=0.04) and is likely a false positive.

5. Locuszoom should be LocusZoom

6. Page 7 – what is the r2 between rs62228062 and rs7291412? Was rs62228062 in the 95% credible set for that locus in Ng et al? And therefore is the predisposition to FS and DD in this region of Chr. 22 likely driven by the same underlying causal variant?

7. Page 7 – how many cases overlapped between FS and DD?

8. Page 7 – I think the misclassification bias discussed above extends to the MR analysis, but the methodology of which cases and controls were used in this analysis is unclear - the methods section needs to be very clearly written.

9. Page 9 – replace elongated with extended.

10. Page 9 – “…chronic condition like DD…” – please read up a bit on DD. It is a fibrotic condition just like FS, it has a similar age of onset (mean 63 years). Many patients with DD also get FS. It is not at all unexpected that both conditions might share some genetic predisposition. For example:

https://doi.org/10.1302/0301-620X.77B5.7559688

https://doi.org/10.1067/mse.2001.112883

11. Page 9, final paragraph, remove the first sentence – it is repetitive, and also misleading, as only Type1 DM was really shown to be causative.

12. Please remove all claims to identifying the “first” variant associated with FS. This should be determined in retrospect, and such claims of primacy are not fitting of a scientific article.

Tables

1. Table 1 should have numbers of participants, not just percentages. I don’t know if all of the cases in UK Biobank were included in this analysis, or just the ones identified by ICD10 coding. It should be the former – if not, please re-run the analyses.

Reviewer #2: The authors conducted a GWAS on frozen shoulder in the UK Biobank sample, and replicated the finding using available GWAS results of a well-powered independent cohort. Using the GWAS results, 1-sample and 2-sample Mendelian randomization analyses were conducted revealing type 1 diabetes but not type 2 diabetes as causal risk factor for frozen shoulder. The analyses were generally sound, the findings robust, and the paper is well written. However, more details regarding the analysis methods need to be provided, and several points listed below should be addressed or clarified.

Although the conducted discovery-replication approach is valid and provides a robust result, it would be of interest to perform additionally a combined UK Biobank and FinnGen GWAS meta-analysis (without replication) to see, if there are any additional genome-wide significant associations on frozen shoulder by maximizing the sample size in the discovery stage.

Methods: please provide how significance was defined in the observational analysis, the MR analysis, the GWAS, and provide the replication criteria applied for the GWAS results.

How was required independence of instruments (SNPs) for the MR assessed?

Results page 6, first paragraph: It’s not quite clear how to “strong” associations were defined and compared, given that the effects presented in Table 1 are based on a mixture of continuous variables on different scales, and binary traits with different prevalence used as exposure. In addition, the sentence “ In a multivariable logistic regression model only the type 1 and type 2 diabetes showed strong association” is imprecise, as also sex had an OR below 1 and a very low adjusted p-value. Please clarify.

Observational results were provided for the duration of diabetes, but the methods are missing describing how diabetes duration (and of which type of diabetes) was assessed. This information needs to be added.

Regarding the 1-sample MR, please provide more details how and on which scale the exposure (i.e. diabetes) was estimated using the genetic factors and included in the second stage, and explain how the causal OR on frozen shoulder can be interpreted with respect to the units of the exposure.

As a sensitivity analysis, I suggest to adjust the first stage association in the 1-sample MR for the traits listed in Table 1 (except diabetes) to correct for potential pleiotropic effects of the SNPs that could lead to invalid instruments.

Please provide the number of instruments included in the respective MR analyses (e.g. in the Supplementary Tables), and provide heterogeneity measures of the 2-sample MR results.

Regarding the Dupuytren’s contracture, are the two WNT7B SNPs rs7291412 and rs62228062 in linkage disequilibrium? Please provide their R².

Although WNT7B is a strong candidate gene as stated in the Discussion, I do not see the conclusion that WNT7B is the likely causal gene at the locus supported by any analysis conducted in this project – the missing eQTL associations and coding variants rather suggest the opposite. Please re-phrase or remove this conclusion.

Please rephrase the “strong” eQTL in the Discussion because the effect size cannot be solely quantified by the association p-value. Furthermore, I encourage to re-assess the use of “strong” in combination with “OR” and “association” throughout the manuscript, and to use more appropriate wordings like higher/lower OR or association p-values.

Discussion end of page 9: the weaker effect of type 2 diabetes on frozen might be because of earlier onset of type 1 diabetes, not because of time point of diagnosis. Please clarify or correct this statement. In the next sentence, it should be written “would have an increased risk” instead of “would be”.

It should be stated were the UK Biobank frozen shoulder GWAS results will be available and accessible.

minor issues:

- page 4: please provide the version of GTEx used

- please add the number of cases and controls to Table 1

- please provide a meaningful p-value for the “T1D no DR3/DR4 haplotyping” meta-analysis results in Supplementary Tables 2 and 3

**Have all data underlying the figures and results presented in the manuscript been provided?**

Reviewer #1: Yes

Reviewer #2: **No: **It is not stated where the UK Biobank frozen shoulder GWAS results are available.

PLOS authors have the option to publish the peer review history of their article (what does this mean?). If published, this will include your full peer review and any attached files.

Reviewer #1: **Yes: **Dominic Furniss

Reviewer #2: No

---

## [Decision Letter · Decision Letter 1]

13 Apr 2021

Dear Dr Weedon,

Thank you very much for submitting your Research Article entitled 'A genome-wide association study of frozen shoulder identifies a common variant of WNT7B and diabetes as causal risk factors' to PLOS Genetics.

The revised manuscript was seen by the original reviewers. As you will see, they are both positive. There are some remaining concerns from reviewer #2 that we ask you address in a hopefully final round of minor revision that will not necessarily need additional external review.

We therefore ask you to modify the manuscript according to the review recommendations. Your revisions should address the specific points made by each reviewer.

[LINK]

Yours sincerely,

Gregory S. Barsh

Editor-in-Chief

PLOS Genetics

Gregory Copenhaver

Editor-in-Chief

PLOS Genetics

Reviewer's Responses to Questions

**Comments to the Authors:**

Reviewer #1: Thank-you for responding so thoroughly to all of the comments from myself and the other reviewer. The paper is now vastly improved, and I strongly recommend publication without any further changes.

Reviewer #2: I thank the authors for addressing all my questions satisfactorily. However, a few additional information need to be added to the revised manuscript:

Please add the effect alleles and their allele frequencies to new Table 3.

As far as I understand, the linear mixed model of BOLD-LMM does not generate estimates that can be directly transformed into odds ratios (compared to e.g. logistic regression). Please add to the methods how the odds ratios (and CI) were approximated from the betas and SEs obtained from the linear mixed model. Are there any known limitations of this approximation e.g. with respect to allele frequency or case-control ratio? Apologies for missing to ask this question after the initial submission already, but addressing this point would be quite important for the reproducibility and interpretation of possible uncertainties of the GWAS results.

On page 6 (MR methods), please specify what the newly added sentence “We also performed Mendelian-Randomisation using betas and odds ratios from the two genome-wide meta-analyses.” exactly means. Did you use for the MR the betas estimated from log(odds ratio) (i.e. taking my comment on the BOLD-LMM results for FS into account)? Given that both exposures and outcome are binary traits, the simple statement “using betas and odds ratios” in the two-sample MR is somehow misleading.

**Have all data underlying the figures and results presented in the manuscript been provided?**

Reviewer #1: Yes

Reviewer #2: Yes

PLOS authors have the option to publish the peer review history of their article (what does this mean?). If published, this will include your full peer review and any attached files.

Reviewer #1: **Yes: **Dominic Furniss

Reviewer #2: No

---

## [Editor Report · Decision Letter 2]

4 May 2021

Dear Dr Weedon,

We are pleased to inform you that your manuscript entitled "A genome-wide association study identifies 5 loci associated with frozen shoulder and implicates diabetes as a causal risk factor" has been editorially accepted for publication in PLOS Genetics. Congratulations!

Yours sincerely,

Gregory Barsh

Editor-in-Chief

PLOS Genetics

Gregory Copenhaver

Editor-in-Chief

PLOS Genetics

Comments from the reviewers (if applicable):

**Data Deposition**

http://datadryad.org/submit?journalID=pgenetics&manu=PGENETICS-D-20-01823R2

**Press Queries**

---

## [Editor Report · Acceptance letter]

21 May 2021

PGENETICS-D-20-01823R2 

A genome-wide association study identifies 5 loci associated with frozen shoulder and implicates diabetes as a causal risk factor 

Dear Dr Weedon, 

We are pleased to inform you that your manuscript entitled "A genome-wide association study identifies 5 loci associated with frozen shoulder and implicates diabetes as a causal risk factor" has been formally accepted for publication in PLOS Genetics! Your manuscript is now with our production department and you will be notified of the publication date in due course.

With kind regards,

Katalin Szabo

PLOS Genetics

On behalf of:
